# Effect of Continuous Renal Replacement Therapy with the oXiris Hemofilter on Critically Ill Patients: A Narrative Review

**DOI:** 10.3390/jcm11226719

**Published:** 2022-11-13

**Authors:** Yupei Li, Peiyan Sun, Kaixi Chang, Mei Yang, Ningyue Deng, Shanshan Chen, Baihai Su

**Affiliations:** 1Department of Nephrology, West China Hospital, Sichuan University, Chengdu 610041, China; 2West China School of Medicine, Sichuan University, Chengdu 610041, China; 3Department of Nephrology, The First People’s Hospital of Shuangliu District, Chengdu 610200, China; 4Med-X Center for Materials, Sichuan University, Chengdu 610041, China; 5Med+ Biomaterial Institute of West China Hospital, Sichuan University, Chengdu 610041, China

**Keywords:** oXiris hemofilter, critical illness, acute kidney injury, immune dysfunction, continuous renal replacement therapy

## Abstract

Critically ill patients with sepsis and severe COVID-19 are commonly characterized by a dysregulated immune response and an acute kidney injury. Continuous renal replacement therapy (CRRT) is now proposed as a promising adjuvant therapy to treat these critically ill patients by removing cytokines, pathogen-associated molecular patterns, and damage-associated molecular patterns from the blood. Although multiple hemofilters, including high-cutoff membranes, the oXiris hemofilter, the CytoSorb hemoadsorption device, and the Toraymyxin hemoperfusion cartridge, have been used in current clinical practice, the use of the oXiris hemofilter in critically ill patients is of particular interest because it is the only kind of hemofilter that can provide renal replacement therapy, remove endotoxins, and adsorb cytokines simultaneously. During the past five years, a growing body of literature has shown that CRRT with the oXiris hemofilter can improve hemodynamics and organ function and can decrease cytokines and endotoxins in both septic and COVID-19 patients. Here, we performed a narrative review to describe the development history of the oXiris hemofilter and to discuss the therapeutic effect of oXiris-CRRT on critically ill patients by searching the PubMed, Web of Science, and clinicaltrials.gov databases for articles published from inception to 8 September 2022 (updated on 1 November) with an English language restriction. We also summarized the current knowledge on anticoagulation techniques and safety concerns when delivering oXiris-CRRT sessions.

## 1. Introduction

Critical illnesses, including sepsis and severe COVID-19, are commonly characterized by a dysregulated immune response and multiple organ dysfunction, such as acute respiratory distress syndrome, acute kidney injuries (AKIs), and neurological dysfunctions, and are associated with high morbidity and mortality [1,2]. Beyond conventional antibiotic therapy and fluid resuscitation therapy, continuous renal replacement therapy (CRRT) has also been widely used in intensive care units (ICU) to provide renal support and to modulate the dysregulated immune response for patients with AKIs and immune dysfunctions worldwide [1]. Traditional CRRT mainly removes solutes and water to maintain hemostasis through diffusion and convection mechanisms using semipermeable hemofilters [3]. However, contemporary CRRT is also proposed as a promising therapy to remove proinflammatory cytokines, pathogen-associated molecular patterns, and damage-associated molecular patterns through the adsorption mechanism in ICU settings [4]. In fact, the solute removal spectrum of a specific hemofilter is significantly dependent on its own membrane/adsorbent structure and treatment dose. In the current clinical practice, multiple hemofilters, including high-cutoff membranes, the oXiris membrane, the CytoSorb hemoadsorption device, and Toraymyxin hemoperfusion cartridges, are used to treat critically ill patients [5]. Among them, the oXiris hemofilter, a high-permeability polyacrylonitrile-based membrane (polyacrylonitrile: AN69), is the only kind of hemofilter that can provide renal replacement therapy, remove endotoxin molecules, and adsorb cytokines simultaneously [6]. Although two previous publications generally discussed the use of oXiris and other hemofilters in patients with sepsis or COVID-19, respectively [7,8], they unfortunately failed to address the safety concerns when delivering the oXiris-CRRT sessions. With the increasing use of oXiris-CRRT in both septic and severe COVID-19 patients during the COVID-19 pandemic, it has become available and important to summarize the latest clinical evidence to investigate the efficacy and safety of oXiris-CRRT in critically ill patients through an updated narrative review that exclusively focuses on the oXiris hemofilter.

Here, we searched the PubMed, Web of Science, and clinicaltrials.gov databases for articles published from inception to 8 September 2022 (updated on 1 November) with an English language restriction. Search terms included “oXiris”, “AN69ST”, “surface-treated AN69 membrane”, “sepsis”, “septic shock”, and “COVID-19”. Clinical studies were eligible for inclusion if they measured the effect of oXiris-CRRT in critically ill patients with sepsis or COVID-19 and if they were cohort studies, case series studies, or randomized controlled trials (RCTs). Preclinical studies that investigated the adsorption kinetics of endotoxins and cytokines of the oXiris hemofilter were also included. We first discussed the developmental history of the oXiris hemofilter and then summarized the most recent literature regarding the clinical use of oXiris-CRRT in sepsis and COVID-19 patients. Finally, the current review focused on the current knowledge on the anticoagulation techniques and safety concerns during CRRT sessions with the oXiris hemofilter.

## 2. History of the oXiris Hemofilter

The oXiris hemofilter (Baxter, Meyzieu, France) employs a positively-charged polyethyleneimine (PEI) coating on an AN69 membrane to achieve simultaneous cytokine adsorption and endotoxin removal from the bloodstream [9]. The conventional AN69 membrane is made of a copolymer of acrylonitrile and sodium methallyl sulfonate, which significantly eliminate cytokines through the negatively-charged sulfonate groups in their bulk area [10]. However, AN69 membranes can interact with the blood to cause bradykinin generation and subsequent anaphylactic reactions in patients being dialyzed with AN69 membranes [11]. In this regard, an AN69 surface-treated (AN69ST) membrane with a PEI coating was developed to improve hemocompatibility by neutralizing the negative charges of conventional AN69 membranes. The PEI layer of the AN69ST membrane can also adsorb heparin to allow heparin priming before use [12]. The oXiris hemofilter further advances the AN69ST membrane by grafting more PEI molecules onto the AN69ST membrane surface to increase the binding of negatively-charged endotoxins with the PEI layer [13]. The oXiris hemofilter is now the only hemofilter that can provide renal replacement therapy, remove endotoxin molecules, and adsorb cytokines simultaneously. Currently, the oXiris hemofilter is available for stand-alone use with different CRRT modalities, such as continuous venovenous hemofiltration, hemodiafiltration, or hemodialysis, via the PrismaFlex and PrisMax systems (Baxter, Deerfield, IL, USA).

## 3. Clinical Use of the oXiris Hemofilter in Sepsis

### 3.1. Rationale

Sepsis, the leading cause of AKIs in critically ill patients, is defined as a dysregulated host response secondary to infection [14,15]. Sepsis has become a major healthcare concern with considerable global economic consequences due to its high incidence and mortality rate for almost four decades [16]. During sepsis, dysregulated immune activation occurs after endotoxins are recognized by pattern-recognition receptors [17]. This signal can further activate leukocytes and induce the production and release of both pro- and anti-inflammatory cytokines, such as tumor necrosis factor-alpha (TNF-α), interleukin-1 (IL-1), IL-6, IL-8, and IL-10. The release of these cytokines into the bloodstream is termed a “cytokine storm”, which further contributes to cellular injury, catabolism, and multiple organ dysfunction [6,18]. In fact, high endotoxin and cytokine levels are associated with multiple organ failure and a high mortality rate in sepsis [19]. Therefore, the removal of endotoxins and cytokines by oXiris-CRRT may have benefits for septic patients with an AKI [15,18].

### 3.2. Preclinical Studies

Rimmelé et al. first compared the effect of the oXiris hemofilter with the AN69 hemofilter on the adsorption kinetics of both endotoxins and cytokines in vitro and in vivo [20]. Under in vitro conditions, 500 mL of bovine serum spiked with 40 EU/mL lipopolysaccharides was circulated in a closed extracorporeal circuit in contact with the oXiris membranes at a flow rate of 250 mL/min. Then, the serum samples were withheld at 0, 10, 30, and 60 min to investigate the adsorption kinetic profile of endotoxins and cytokines using the oXiris hemofilter. The results showed that 66% of endotoxins were adsorbed by the oXiris hemofilter after the 60 min in vitro hemofiltration experiment, while the AN69 hemofilter had little effect on endotoxin removal. Meanwhile, the oXiris hemofilter also showed a greater cytokine adsorption capacity than the AN69 hemofilter. In a porcine sepsis model induced by live *Pseudomonas aeruginosa*, the authors further demonstrated that CRRT with the oXiris hemofilter significantly improved hemodynamics, as evidenced by lower fluid requirements, lactate levels, and pulmonary arterial hypertension compared to the AN69 hemofilter. Additionally, a significant decrease in the endotoxin level after 1 h of hemofiltration was observed in the oXiris group [20]. More recently, another in vitro study by Malard et al. compared the adsorption efficiency of inflammatory mediators and endotoxins of the oXiris hemofilter (Baxter, Meyzieu, France), the CytoSorb hemofilter (CytoSorbents Corporation, New Jersey, USA), and the Toraymyxin hemofilter (Toray Industries, Tokyo, Japan) [21]. Heparinized human plasma was preincubated with pathological quantities of inflammatory cytokines or endotoxins and then filtered in a closed-loop circulation model for 2 h. The results showed that there was no significant difference in the endotoxin removal rates between the oXiris hemofilter (68.0 ± 4.4%) and the Toraymyxin hemofilter (83.4 ± 3.8%) at 120 min, and the oXiris hemofilter displayed an adsorption efficiency similar to the CytoSorb hemofilter for the removal of most cytokines [21]. Specifically, the total endotoxin adsorption amount over 6 h with oXiris was 6.9 vs. 9.7 μg for Toraymyxin. Considering that the endotoxin plasma load in the blood of septic patients was previously reported to be in the range of 3–30 μg [22], these results supported the clinically relevant endotoxin removal capacities of the oXiris^®^ hemofilter.

### 3.3. Clinical Studies

Encouraging results from in vitro studies have significantly increased the clinical use of the oXiris^®^ hemofilter in septic patients in Europe and the Asia–Pacific region. In a randomized crossover double-blind study enrolling 16 sepsis-associated AKI patients who had an endotoxin level higher than 0.03 EU/mL, Broman et al. found that CRRT with the oXiris hemofilter was associated with a greater removal of endotoxins and cytokines (namely, TNF-α, IL-6, IL-8, and IFN-γ) than that with a standard AN69ST hemofilter (Baxter, Meyzieu, France) [13]. Notably, endotoxin levels decreased during the CRRT treatment in seven of nine (77.8%) patients in the oXiris group but only in one of six (16.7%) patients in the AN69ST group, suggesting that oXiris-CRRT can effectively eliminate endotoxins and cytokines from the bloodstream of septic patients in real clinical practices.

Dating back to 2013, Shum et al. first conducted a retrospective case-series study to determine whether the use of the oXiris hemofilter improved organ dysfunction in patients with severe Gram-negative bacterial infections. A total of 30 patients with sepsis-associated AKIs who received one oXiris-CRRT session were recruited [23]. The results showed that the SOFA score was significantly reduced by 37% at 48 h post-initiation of CRRT with the oXiris hemofilter versus an increment of 3% in the historical controls that received CRRT with a polysulfone hemofilter. Meanwhile, there were no significant differences in the ICU and in-hospital mortality rates between the two groups. Subsequently, a growing number of small-size clinical studies collectively found that CRRT with the oXiris hemofilter could improve hemodynamics and decrease the cytokines, procalcitonin, endotoxins, and SOFA scores in septic patients with AKIs [24,25,26,27,28,29]. Additionally, Schwindenhammer et al. found that oXiris-CRRT was associated with a higher observed survival than was predicted by the SAPS II score in patients with septic shock in two French centers [30].

Most recently, the conflicting results of several cohort studies that compared the effects of oXiris vs. AN69ST hemofilters on cytokine levels and clinical outcomes in Chinese septic patients have become available. Zang et al. found that patients receiving oXiris-CRRT had a more remarkable improvement in hemodynamics and had lower cytokine concentrations than those receiving AN69ST-CRRT, but there were no significant differences in clinical outcomes, such as the in-hospital mortality rate, intensive care unit length of stay (LOS), and hospital LOS [31]. In another retrospective observational study enrolling 136 patients with sepsis and AKIs, Guan et al. reported a significantly lower 7-day (47.1% vs. 74.2%) and 14-day mortality rate in the oXiris group than in the AN69ST group (58.5% vs. 80.3%), although the difference in the 90-day mortality rate (71.4% vs. 81.8%) was insignificant [6]. Compared with AN69ST-CRRT, oXiris-CRRT was also associated with a faster reduction in the SOFA score and a greater decrease in the procalcitonin level and vasoactive-inotropic score. Furthermore, the authors found that oXiris-CRRT was associated with a lower risk of death with a hazard ratio of 0.500 (95% CI: 0.280–0.892; *p* = 0.019) than AN69ST using a multivariate Cox regression model. Likewise, Xie et al. performed an inverse probability on the treatment-weighted analysis to compare the effect of oXiris-CRRT vs. ANS6ST-CRRT on patient-centered clinical outcomes, including the 28-day mortality rate, the 72 h lactate level, and the need for norepinephrine [32]. Their results showed that oXiris-CRRT was associated with a lower 28-day mortality rate (47.3% vs. 73.3%) and reduced lactate levels, norepinephrine doses, and procalcitonin in septic shock patients vs. AN69ST-CRRT. It is also noteworthy that these conflicting results should be interpreted with caution because of the inherent limitations of cohort studies, such as selection bias and small sample size. In fact, the mortality rate of septic patients in the oXiris group remained high (in a range of 40.9% to 71.4%) [6,31,32], suggesting that oXiris-CRRT was usually used as a remedial treatment in the practice at that time. Therefore, further studies should be performed to investigate whether the early initiation of oXiris-CRRT may be beneficial in the reduction of in-hospital mortality rates in patients with sepsis-associated AKIs. Meanwhile, it is also important to identify whether the oXiris hemofilter could reduce the mortality rate vs. the standard of care and other hemofilters, such as the Cytosorb hemofilter and the Toraymyxin hemoperfusion cartridge. The main findings of the clinical trials that evaluated the effect of oXiris-CRRT on septic patients are further summarized in Table 1.

Although there have been a handful of encouraging clinical studies as mentioned above, the absence of high-quality studies and clinical guidelines makes the wide use of oXiris not yet advocated, leading to variability in clinical practice. Currently, the delivery of oXiris-CRRT is mainly based on consensus recommendations from Europe and the Asian–Pacific region (see Figure 1) [9,33]. First, the initiation of oXiris-CRRT is recommended when patients develop both an AKI and septic shock. The decision to initiate oXiris-CRRT is usually driven by a clinical judgment based on a combination of clinical signs, such as hemodynamic instability; microcirculatory dysfunction; MODS; and laboratory parameters, such as elevated serum levels of procalcitonin, IL-6, and lactate [9]. However, the setting of AKIs pertains to patient selection, and the optimal circumstances for CRRT initiation remain one of the most controversial and clinically uncertain aspects of CRRT delivery. The initiation of dialysis early versus delayed in the intensive care unit (IDEAL-ICU) trial in 2018 that enrolled 488 patients with severe AKIs due to septic shock and with no urgent indications for RRT initiation found that early RRT initiation, as compared to delaying RRT initiation by 48 h, did not reduce the 90-day mortality rate [34]. Likewise, in 2020, the multinational standard versus accelerated renal replacement therapy in acute kidney injury trial also showed that early RRT initiation did not improve the primary outcome of the 90-day all-cause mortality rate, but it increased the likelihood of persistent RRT dependence at 90 days after randomization. As such, we also believe that the timing of oXiris-CRRT initiation (early or late start) certainly has an impact on the clinical outcomes of patients. Second, the oXiris hemofilter is recommended to be changed at 12 to 24 h post-initiation if patients have persistently high cytokine levels or if premature circuit clotting is anticipated. However, the filter lifespan may be up to 72 h if patients show remarkable improvements in clinical signs and laboratory parameters after oXiris-CRRT [9]. Third, a therapeutic effect evaluation should be performed at 24 h post-initiation of oXiris-CRRT to determine whether the oXiris treatment succeeded or failed based on clinical observations and laboratory markers. Fourth, the CRRT prescriptions for the oXiris hemofilters do not differ significantly from other conventional hemofilters. Regional citrate anticoagulation (RCA) or systemic heparin anticoagulation is recommended. As for CRRT intensity, the landmark VA-NIH ATN and RENAL trials collectively demonstrated that intensive small-solute clearance was not superior to less-intensive clearance [35,36]. Accordingly, an effluent flow of 20–25 mL/kg/h has become the standard of care for the delivery of CRRT [3]. Likewise, the consensus recommendations recommend that the target dose of an oXiris-CRRT session should reach 20 to 35 mL/kg/h with a blood flow rate of 150 to 200 mL/min [9]. Of note, the key questions in delivering oXiris-CRRT remain uncovered, such as the timing of initiation and weaning, recommended therapeutic dose, optimal duration of use, indication of filter exchange, and definition of treatment success, which, unfortunately, limits the wide use of the oXiris hemofilter. Moreover, despite currently having insufficient information to generate a comprehensive cost-effectiveness analysis of oXiris-CRRT, a recent retrospective study at our institution found that the cost of oXiris-CRRT is about two times more than that of AN69ST-CRRT, with an average cost of 1800 dollars per filter. Therefore, the use of oXiris in low-income countries may be hindered by its high cost, the limited availability and reliability of endotoxin detection experiments, and restricted access to oXiris. High-quality RCTs or prospective randomized crossover studies are desperately needed for better clinical decisions.

## 4. Clinical Use of the oXiris Hemofilter in COVID-19

### 4.1. Rationale

Multiple studies have reported that cytokine storms, characterized by highly elevated levels of proinflammatory cytokines such as interleukin-6 (IL-6), IL-1β, IL-18, and granulocyte–macrophage colony-stimulating factor, and immunothrombosis are associated with the disease severities and clinical outcomes of critically ill COVID-19 patients [37]. Recently, cytokine storms have been proposed to crosstalk with immunothrombosis and endothelial dysfunctions to mediate thrombotic events, multiple organ dysfunctions, and death in COVID-19 patients [38]. A recent meta-analysis identified an overall estimated pooled incidence of venous thromboembolisms of 17.0 % in hospitalized patients with COVID-19 [39]. Beyond thromboembolisms, AKIs are another common complication of severe COVID-19, with up to 45% of COVID-19 patients in the ICU setting requiring RRT [38]. The pathophysiology of COVID-19 AKIs is thought to involve local and systemic inflammatory and immune responses, endothelial injuries, the activation of coagulation pathways and the renin–angiotensin system, ischemic acute tubular necrosis, and the direct viral invasion of renal proximal tubular cells and podocytes [38,40]. In a multicenter cohort study of 3099 critically ill adults with COVID-19 across the USA, Gupta et al. found that AKIs treated with kidney replacement therapy (AKI-RRT) were associated with a hospital mortality rate of >60% and that patient-level risk factors for AKI-RRT included chronic kidney disease, male gender, non-White race, hypertension, diabetes mellitus, being overweight, a higher d-dimer, and a greater severity of hypoxemia on ICU admission [40]. Owing to its capacity to adsorb proinflammatory mediators from the bloodstream during CRRT sessions, the oXiris hemofilter was authorized for emergency use to overcome AKIs and/or cytokine storms in adults with COVID-19 by the FDA in April 2020.

### 4.2. Clinical Evidence

Early small-size case series collectively found that CRRT with the oXiris hemofilter significantly decreased proinflammatory cytokine levels and improved hemodynamics and organ function in critically ill COVID-19 patients [41,42,43,44,45,46]. Compared to the mortality rates calculated by the acute physiology and chronic health evaluation IV score, the mean observed mortality rates were lower after oXiris-CRRT treatment [44,46]. Premužić et al. also demonstrated that critically ill COVID-19 patients receiving the oXiris treatment survived significantly longer than other ICU COVID-19 patients [44]. In contrast, another single-center study reported negative results for removing circulating cytokines and inflammatory chemokines in non-AKI patients with severe COVID-19, which might be attributed to the relatively lower concentration of IL-6 in COVID-19 patients than in patients with septic shock [47]. Furthermore, the differences in inflammatory subphenotypes, the SARS-CoV-2 viral burden, and comorbidities may also have an impact on the production and release of circulating cytokines and chemokines [47]. These findings suggested that the routine clinical use of the oXiris membrane in non-AKI COVID-19 patients should be considered with caution. The main findings of the clinical trials that evaluated the effect of oXiris-CRRT on COVID-19 patients are further summarized in Table 2.

Currently, there is an ongoing open-label RCT (oXAKI-COV study) comparing CRRT with the oXiris membrane vs. the standard AN69 membrane during a 72 h treatment period in critically ill COVID-19 patients with AKIs (NCT04597034) [48]. The primary outcome of the oXAKI-COV study was the change in the norepinephrine requirement by at least 0.1 mg/kg/min to maintain a similar mean arterial pressure after the initiation of CRRT. Secondary outcome measures included the change in serum IL-6, IL-10, and TNF-α levels and the change in the length of ICU stays in these patients. It is believed that the final analysis of such a high-quality RCT could provide solid evidence for better clinical decisions.

## 5. Anticoagulation Strategy and Drug Clearance during oXiris-CRRT Sessions

### 5.1. Anticoagulation Strategy

Membrane clotting is a significant pitfall in CRRT and is potentially associated with severe clinical consequences in critically ill patients who have already been in a hypercoagulable state. Consequently, adequate systemic heparin anticoagulation or RCA is critically required to maintain the patency of the extracorporeal circuit and to preserve the extracorporeal transmembrane clearance of solutes during CRRT sessions. Unlike conventional polysulfone or polymethyl methacrylate hemofilters, the oXiris^®^ hemofilter is pretreated with heparin during its manufacturing process, ensuring a high concentration (4500 ± 1500 IU/m^2^) of heparin and theoretically removing the need for heparin priming. However, the coagulation of the extracorporeal circuit could be the most common adverse effect in patients receiving oXiris-CRRT [29], and the consensus recommendations from the Asian–Pacific region recommend the use of heparin or citrate anticoagulation during oXiris-CRRT sessions [9]. In 2021, Wong et al. conducted a pilot randomized controlled trial to compare the filter lifespan of the oXiris vs. the AN69 membranes in 20 critically ill patients at high bleeding risk who underwent nonanticoagulation CRRT. All circuits were first primed with 1 L of heparin saline (5000 IU) to allow heparin priming and were then primed with an additional 1 L of normal saline to flush out the residual heparin before connecting them to the patients. The results showed that the median filter lives for oXiris hemofilters were insignificantly shorter than those for conventional AN69 hemofilters (13 h vs. 18 h, *p* = 0.10) [49]. In addition, the transmembrane pressures in the oXiris filter circuits were higher than those in the AN69 filter circuits by 12 h, suggesting faster membrane clogging with oXiris. This phenomenon is in contrast to the experience of systemic heparin reduction along with the use of the AN69ST membrane in chronic hemodialysis, which has a much shorter treatment duration (usually 4 h). In another recent multicenter prospective study enrolling 97 septic patients undergoing oXiris-CRRT, Villa et al. observed that premature clotting occurred in 18 (18.6%) patients, which was relatively low compared to a previously reported value (54%) [50]. A further multivariate logistic regression analysis showed that systemic heparin anticoagulation and RCA were both protective factors in circuit clotting when compared to the nonanticoagulation strategy. It should also be noted that a significantly higher incidence of metabolic alkalosis and hypercalcemia and more premature clotting in hemofilters in COVID-19 patients receiving CRRT with RCA compared to non-COVID-19 patients were observed [51], which reminds us that close monitoring of the acid–base balance appears to be warranted in severe COVID-19 patients receiving oXiris-CRRT with RCA. In conclusion, anticoagulant use is generally needed to avoid premature clotting when delivering oXiris-CRRT in a specific subgroup of critically ill patients with signs of systemic inflammation and with a high risk of filter clotting, such as severe COVID-19 patients.

### 5.2. Drug Clearance

Although current evidence collectively suggests that the use of the oXiris hemofilter in critically ill patients is well tolerated in most cases, unwanted drug clearance by the oXiris membrane through either diafiltration or adsorption remains a major safety concern. An enhanced clearance of antibiotics may lead to a decrease in drug concentrations in the blood to subtherapeutic levels, making it impossible to successfully treat infections or septic shock. In a recent study evaluating the adsorption of vancomycin in CRRT circuits, there was a significantly greater adsorption of vancomycin in 2 h by AN69ST vs. the polysulfone membrane [52]. The adsorption of antibiotics on the AN69ST membrane reached 181.88 mg. Honore et al. therefore recommended that daily vancomycin maintenance doses close to 3000 mg were needed during the first 3 days of AN69ST-CRRT treatment [53]. Accordingly, despite the lack of solid evidence, it should still be kept in mind that the dose correction of antibiotics is crucial when delivering oXiris-CRRT in critically ill patients because the oXiris hemofilter shares a structure and a diafiltration property similar to those of the AN69ST hemofilter in nature. Further investigations into the drug clearance kinetics during oXiris-CRRT in real clinical practices are recommended.

Although the oXiris hemofilter exhibited a different membrane structure and solute clearance spectrum than other conventional polysulfone or polymethyl methacrylate hemofilters, the prescription parameters, such as the target effluent dose, blood flow, and anticoagulation regimens for oXiris-CRRT, do not differ significantly from those for CRRT with conventional hemofilters in the current practice [9]. Beyond membrane clotting and drug clearance, a broad range of severe adverse effects, such as the loss of nutrients, hypophosphatemia, heat loss, and vascular-access-related complications, can also occur during CRRT sessions. In 2012, Maynar et al. innovatively introduced a dialytrauma concept that encompassed all the harmful adverse events related to CRRT while providing a framework for the early recognition and prevention of these events [54]. The authors further developed a dialytrauma checklist to offer a stepwise approach before starting CRRT in an individual patient [54]. Therefore, we believe it is also of great significance to develop a specific dialytrauma checklist for oXiris-CRRT when more clinical evidence becomes available to enable the timely tailoring of CRRT prescriptions in a safe and efficient manner.

## 6. Ongoing Clinical Studies Evaluating oXiris-CRRT in Critically Ill Patients

Currently, there are also a large number of ongoing clinical trials that aim to evaluate the safety and efficacy of oXiris-CRRT in critically ill patients with sepsis-associated AKIs (NCT05575024, NCT04952714, NCT04997421) or with a post cardiac surgery (NCT05182723, NCT04201119). Furthermore, researchers can also participate in the Global ARRT International Registry (NCT03807414, official website: www.arrt.eu, accessed on 9 September 2022) to accelerate the generation of better clinical evidence in this field. Despite the differences in the patient populations, study designs, grouping, and primary outcomes across these studies, we believe that the final analysis of these clinical trials will certainly sharpen the daily practice of oXiris-CRRT in the near future. The detailed information of ongoing clinical studies evaluating oXiris-CRRT in critically ill patients is further summarized in Table 3.

## 7. Conclusions and Outlook

In conclusion, the oXiris hemofilter functions through its unique three-layer membrane structure to remove endotoxins, eliminate cytokines, and provide renal replacement for critically ill patients with AKIs and immune dysfunctions. In vitro studies collectively showed that the oXiris hemofilter had sufficient endotoxin and cytokine adsorption capacities to decrease the lethal endotoxin and cytokine burdens in the blood of septic patients. Moreover, many small-size clinical studies found that oXiris-CRRT was associated with improved hemodynamics and organ function and decreased cytokine levels, procalcitonin, and endotoxin burdens in both septic and severe COVID-19 patients. However, the effect of CRRT with the oXiris hemofilter on patient-centered outcomes, such as the length of hospital stay and the mortality rate, remains inconclusive. It should also be noted that anticoagulant use is usually required to prevent extracorporeal circuits from clotting during oXiris-CRRT sessions.

Despite the increasing use of the oXiris hemofilter in the intensive care setting, the absence of high-quality studies and clinical guidelines is responsible for the lack of a standard oXiris-CRRT prescription in clinical practices. In fact, a set of key questions in delivering oXiris-CRRT in patients with critical illnesses remains unknown, such as the timing of initiation and weaning, the recommended therapeutic dose, the optimal duration of use, the indication of filter exchange, and the definition of treatment success. Meanwhile, the therapeutical effect of the oXiris hemofilter should also be compared with other hemofilters, such as the Cytosorb hemofilter and the Toraymyxin hemoperfusion cartridge, beyond its maternal AN69ST membrane in patients with sepsis in the future. It is also noteworthy that critically ill patients with sepsis or COVID-19 represent significantly heterogeneous disease severities and patient characteristics, making it important to identify a specific patient subgroup that will most likely benefit most from oXiris-CRRT treatment. Another important issue when designing clinical trials that evaluate the efficacy of the extracorporeal blood purification in critically ill patients is the selection of the study endpoint. The choice of the mortality rate as an endpoint for studies in the ICU setting is potentially fraught with problems. Indeed, a failure to show a survival effect by a certain therapeutic regimen does not simply mean that the regimen is worthless in the ICU setting, and care must be taken to avoid potentially useful interventions being abandoned through the limitations of patient selection within the clinical trial format [55]. Other patient-centered outcomes, such as ventilator-free days, the avoidance of chronic kidney disease, and long-term cognitive function, can be considered as alternative endpoints. Regardless, the final analysis of the ongoing Global ARRT International Registry study (NCT03807414) and other randomized controlled trials will certainly sharpen the clinical practice of oXiris-CRRT in the near future.

## Figures and Tables

**Figure 1 jcm-11-06719-f001:**
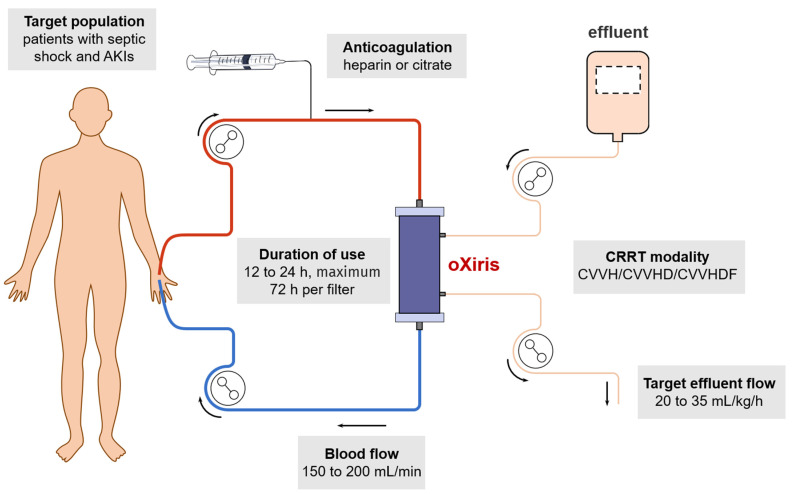
Key recommendations for the delivery of oXiris-CRRT by the Asia–Pacific expert consensus [9].

**Table 1 jcm-11-06719-t001:** Summary of main clinical trials evaluating oXiris-CRRT in septic patients.

Author	Publication Year	Country	Study Design	Sample Size	Type of Infection	Reference Hemofilter	Main Findings
Shum [23]	2013	Hong Kong, China	retrospective case series with historical controls	30	Gram-negative bacterial infection	polysulfone high-flux hemofilter	The SOFA score was significantly reduced by 37% at the 48 h post-initiation of oXiris-CRRT with no significant differences in the ICU and in-hospital mortality rates.
Turani [24]	2019	Italy	retrospective case series	60	Gram-negative bacterial infection (60%), Gram-positive bacterial infection (35%), and fungal infection (5%)	NA	oXiris-CRRT might improve the oxygenation index and hemodynamics and decrease the cytokines, procalcitonin, endotoxin burden, and SOFA scores in septic patients with AKIs.
Broman [13]	2019	Sweden	prospective randomized crossover study	16	Confirmed or suspected Gram-negative bacterial infection	AN69ST membrane	oXiris-CRRT was associated with the effective removal of endotoxins and TNF-α, IL-6, IL-8, and IFN-γ in patients with septic-shock-associated AKIs and with a beneficial effect on hemodynamics.
Schwindenhammer [30]	2019	France	retrospective cohort study	31	Gram-negative bacterial infection (50%)	NA	oXiris-CRRT resulted in a higher observed survival than predicted by the SAPS II score in patients with septic shock. Hemodynamics and lactatemia appeared to improve, especially in patients with intra-abdominal sepsis and Gram-negative bacterial infections.
Zhang [28]	2019	China	retrospective case series	6	NA	NA	In patients with septic shock, oXiris-CRRT significantly decreased the heart ratio, IL-6, and CRP levels, and improved the hemodynamic status.
Lumlertgul [25]	2021	Japan	retrospective case series	35	Gram-negative bacterial infection (77.1%), Gram-positive bacterial infection (8.6%), and fungal infection (8.6%)	NA	In patients with septic shock, oXiris-CRRT was associated with an improvement in the hemodynamic status as evidenced by the increased mean arterial pressure and decreased norepinephrine dose. Cardiovascular SOFA scores significantly decreased over 72 h.
Zhai [26]	2021	China	retrospective observational study	53	NA	NA	Compared with conventional CRRT, oXiris-CRRT effectively improved hemodynamics; reduced serum IL-6, IL-10, and endotoxin levels; and shortened the ICU-LOS and organ support duration in septic patients.
Zang [31]	2022	China	prospective observational study	44	Gram-negative bacterial infection (75%), Gram-positive bacterial infection (11.4%), and fungal infection (6.8%)	AN69ST membrane	Compared with AN69ST-CRRT, patients treated with oXiris-CRRT had a more remarkable improvement in hemodynamics and lower cytokine levels. However, there were no differences in the in-hospital mortality rate, ICU-LOS, and hospital LOS.
Xie [32]	2022	China	retrospective cohort study	76	Gram-negative bacterial infection (72.4%), Gram-positive bacterial infection (15.8%), and fungal infection (55.3%)	AN69ST membrane	Compared with AN69ST-CRRT, oXiris-CRRT was associated with a lower mortality rate and reduced lactate levels, norepinephrine doses, and procalcitonin in patients with septic shock.
Guan [6]	2022	China	retrospective observational study	136	Confirmed or suspected Gram-negative bacterial infection	AN69ST membrane	Compared with AN69ST-CRRT, oXiris-CRRT was associated with a faster reduction in the SOFA score, a greater decrease in procalcitonin, and a lower short-term (7-day and14-day) mortality rate in septic shock patients with AKIs despite no difference being found in the 90-day mortality rate between the two groups.
Feng [27]	2022	China	randomized controlled trial	16	Confirmed or suspected Gram-negative bacterial infection	AN69ST membrane	Compared with AN69ST-CRRT, oXiris-CRRT was significantly associated with lower procalcitonin and IL-6 concentrations and improved hemodynamics.
Zhou [29]	2022	China	retrospective observational study	90	Gram-negative bacterial infection (50%), Gram-positive bacterial infection (18.9%), and fungal infection (13.3%)	NA	oXiris-CRRT was associated with decreased procalcitonin and IL-6 concentrations, a reduced SOFA score, and improved hemodynamics.

Abbreviations: SOFA represents sequential organ failure assessment; CRRT represents continuous renal replacement therapy; and LOS represents length of stay.

**Table 2 jcm-11-06719-t002:** Summary of main clinical trials evaluating oXiris-CRRT in COVID-19 patients.

Author	Publication Year	Country	Study Design	Sample Size	Main Findings
Padala [41]	2020	USA	case series	3	oXiris-CRRT decreased the levels of IL-6, the erythrocyte sedimentation rate, and CRP in critically ill patients with COVID-19.
Zhang [42]	2020	China	case series	5	Oxiris-CRRT might reduce the level of IL-6, IL-8, and CRP and improve the hemodynamic status as well as organ function.
Villa [46]	2020	Italy	prospective observational study	37	oXiris-CRRT was associated with a reduction in the serum IL-6 level, an attenuation in systemic inflammation, an improvement in multiorgan dysfunctions, and a decrease in the expected ICU mortality rate.
Rosalia [43]	2022	North Macedonia	prospective cohort study	44	oXiris-CRRT is associated with a reduction in ferritin, CRP, fibrinogen, and IL-6, and a resolution in numerous cytopenias.
Premužić [44]	2022	Zagreb Croatia	retrospective observational study	15	oXiris-CRRT provides a significant reduction in IL-6, an improvement in the respiratory status, and a reduction in SOFA score severity.
Ugurov [45]	2022	North Macedonia	case series	15	The combination of systemic heparin anticoagulation regimens and oXiris-CRRT might reduce hyperinflammation, prevent coagulopathy, and support clinical recovery.
Kang [47]	2022	China	prospective nonblind randomized controlled study	17	oXiris-CRRT showed no advantage in removing circulating cytokines and inflammatory chemokines in non-AKI patients with severe and critical COVID-19.

Abbreviations: CRRT represents continuous renal replacement therapy; CRP represents C-reactive protein; and IL represents interleukin.

**Table 3 jcm-11-06719-t003:** Summary of ongoing clinical studies evaluating oXiris-CRRT in critically ill patients.

NCT Number	Study Title	Status	Location	Population	Grouping	Study Type	Estimated Enrollment	Key Primary Outcome	Estimated Completion Date
NCT05575024	Real-time Monitoring of Hemodynamic Stability in Septic Shock Patients Undergoing Continuous Renal Replacement Therapy: Propensity Score-matched Comparison Between oXiris and Polysulfone Membranes	Not yet recruiting	South Korea	Sepsis-associated AKI	Experimental: oXiris Control: polysulfone	Observational	98	28-day all-cause mortality rate	31 December 2024
NCT04952714	Renal Replacement Therapy with a Cytokine Absorption Filter (oXiris ^®^) in Patients with Septic Shock: A Case-control Study Nested in a Cohort	Not yet recruiting	NA	Sepsis-associated AKI	oXiris	Observational	93	28-day mortality rate	December 2022
NCT04033029	Antibiotic Plasma Concentrations During Continuous Renal Replacement Therapy with a High Adsorption Membrane (oXiris^®^)	Recruiting	Spain	Sepsis	oXiris	Observational	50	Time above MIC for beta-lactams, and total drug AUC24/MIC ≥ 666 for daptomycin	31 December 2023
NCT03426943	Endotoxins and Cytokines Removal During Continuous Hemofiltration With oXiris™	Recruiting	France	Septic shock and peritonitis	Experimental: oXiris Control: HF1400 filter	Interventional	50	Change in IL-6 and endotoxin concentration	21 March 2023
NCT04886180	Evaluation of Oxiris Membrane as a Treatment for Ischemia-reperfusion Syndrome in Cardiogenic Shock Treated with Extracorporeal Life Support (ECMO/ECLS): A Randomized Pilot Study ECMORIX	Recruiting	France	Cardiogenic shock	Experimental: oXiris Control: AN69ST membrane	Interventional	40	Change in plasma endotoxin concentration	June 2024
NCT04997421	Safety and Efficacy of HA380 Hemoadsorption in Combination with oXiris Membrane for Continuous Hemodiafiltration in Patients with Septic Shock—HEMOX-HDF Trial	Recruiting	Finland	Septic shock and AKI	Experimental: oXiris + HA330 Control: oXiris	Interventional	40	Intensive care mortality rate	December 2025
NCT05182723	Evaluation of the Effectiveness of Extracorporeal Methods for Removing Mediators of Systemic Inflammation in Patient with Multiple Organ Dysfunction Syndrome After Heart and Aortic Surgery	Recruiting	Russia	MODS post cardiac surgery	Experimental: oXiris + HA330 Control: oXiris	Interventional	20	Change in CRP, IL-6, and SOFA	31 May 2025
NCT04957316	Effects of Removal of Endotoxin, Cytokines, and Uremic Toxins Using Blood Purification on Patients with Severe Septic Shock	Recruiting	Taiwan	Septic shock	Experimental: oXiris + standard of care Control: standard of care	Interventional	120	Duration of vasopressor and inotrope infusion	December 2024
NCT05595239	The Efficacy of Different Mode of Blood Purification in Septic Children	Not yet recruiting	China	Sepsis	Experimental: oXiris + standard of care Control: standard of care	Interventional	150	Change in cytokine concentration	30 June 2025
NCT03807414	Development of a Web-based Multicenter Registry on the Use of oXiris Membrane for Extracorporeal Blood Purification Therapies in Critically Ill Patients	Recruiting	International	Critical illness	oXiris	Registry	270	NA	30 September 2022
NCT04201119	Effect of Oxiris^®^ Membrane on Microcirculation Following Cardiac Surgery Under Cardiopulmonary Bypass: A Pilot Prospective Monocentric Study (Oxicard Study)	Recruiting	France	Post cardiac surgery	Experimental: oXiris + standard of care Control: standard of care	Interventional	70	Improvement in microcirculatory flow on day 1	30 March 2022

Abbreviations: MIC represents minimum inhibitory concentration; CRP represents C-reactive protein; SOFA represents sequential organ failure assessment.

## Data Availability

Not applicable.

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
