# Peer review of "Effect of Continuous Renal Replacement Therapy with the oXiris Hemofilter on Critically Ill Patients: A Narrative Review"

_jcm, 2022, doi:10.3390/jcm11226719_

Round 1
Reviewer 1 Report
Fair review.
I would like to see an information about cost-effectiveness if any exist or dialitrauma.
Author Response
Thank you for your scientific comments and constructive suggestions to further improve the quality of our manuscript. Unfortunately, there is insufficient information to generate a comprehensive cost-effectiveness analysis of oXiris-CRRT against conventional hemofilters. A recent retrospective study at our institution (Doi: 10.3389/fmed.2022.78962) found that the cost of oXiris-CRRT is about two times more than that of AN69ST-CRRT despite there was no significant difference in hospital expenses between the two groups. We have added a discussion of this issue and call for further investigations to address this knowledge gap. (Line 228-231) For dialytrama, we thank the reviewer for pointing out the newly proposed concept that encompasses all harmful adverse events related to CRRT. However, it should be noted that the use of oXiris-CRRT is still limited and infrequent in most medical centers compared with other conventional hemofilters, and we thus lack sufficient data to prepare a dialytrauma checklist that is exclusively applied to oXiris-CRRT. The expert recommendations based on the Asia Pacific experience also stated that “The parameters for administering the oXiris filter do not differ significantly from those used for CRRT with conventional dialysis filters.” Therefore, we have added a discussion on the dialytrauma concept by introducing the reader with a previous dialytrauma checklist by Moliner et al. (Doi: 10.1159/000342064), rather than creating a new one. (Line 345-358)

Reviewer 2 Report
The article is valuable in its effort to make a summary of the state of art in a field lacking solid evidences derived from randomized controlled trials. Nevertheless there are some issues that have to be addressed.
Major concerns
First of all it would be better to define the type of review selected by authors: narrative review? systematic review? In the second case it could be important to follow the PRISMA guidelines and to revise all the manuscript accordingly. However, even if the selected type is that of narrative review please add the criteria for article selection (year of publication, type of article, content of the article, the databases consulted, etc) because the definition “the most recent literature” is too general. In fact the authors reported articles from 2013 to 2022. Are there any kind of articles excluded from the review? Why? Please justify the exclusion or add the remaining articles.
Please remove from the review the articles available only in Chinese because they are not readable for all readers. It is important that the articles used for the review are widely available for consultation.
Please add a table which summarizes the studies of oXiris in Covid19 patients (like what has already been done in table 1 for studies in septic patients).
In the introduction lines 51 and 52 the authors stated that “the therapeutic effect of oXiris-CRRT … has never been summarized in previous publication”: this is not fully correct. In literature there are two reviews available on Pubmed (Renal Replacement Techniques in Septic Shock. Hellman T, Uusalo P, Järvisalo MJ. Int J Mol Sci. 2021 Sep 23;22(19):10238. doi: 10.3390/ijms221910238. PMID: 34638575 and A Review of Extracorporeal Blood Purification Techniques for the Treatment of Critically Ill Coronavirus Disease 2019 Patients. Niazi NS, Nassar TI, Stewart IJ, Honore PM, Sharma K, Chung KK. ASAIO J. 2022 Apr 12. doi: 10.1097/MAT.0000000000001761. Online ahead of print. PMID: 35417433) that summarize the state of art in patients with sepsis and with Covid19 and cite the filter and briefly reported the studies available. I think it would be better to add these reviews in bibliography and to correct the statement in the introduction (for example: it is the first to focus on oXiris while others…)
Table 1 summarizes the studies on septic patients: I think that it could be improve by adding two/three columns, one for the type of infection that affect patients in the studies (for example: gram negative or gram positive infection, endocarditis, septic shock from … and so on) and one or two with the indication of the filters being compared in the study (if there is a comparison). Than summarize the statements in the “main findings” removing the data of the other columns.
The majority of studies in table 1 shows oXiris filter being compared with AN69ST: are there any study comparing oXiris with CytoSorb for example or other hemofilters? Why? Why not?
In the conclusion the authors stated “international multicentred registry studies”: how many registry studies are ongoing? Where? It would be valuable to give some information about them to allow other researchers to join them.
In the conclusion the authors stated that randomized controlled trials are desperately needed and they mention only an ongoing trial in the article (line 227-228). However, on Clinicaltrials.gov there are many different studies ongoing on oXiris filter: it would be interesting to add a paragraph, before conclusion, that summarize those studies, their aims, the expected term and findings etc., so giving the reader an idea about what is in progress.
Minor reviews:
Table 1: French is a language, the country is France: please amend.
Author Response
Comment 1: The article is valuable in its effort to make a summary of the state of art in a field lacking solid evidences derived from randomized controlled trials. Nevertheless, there are some issues that have to be addressed.
Response: Thank you for your scientific comments and constructive suggestions to further improve the quality of our manuscript. The manuscript has now been thoroughly revised according to the comments from you and the other reviewers. Please find below a list of our detailed responses to your comments and corresponding changes in the revised manuscript.
Comment 2: First of all, it would be better to define the type of review selected by authors: narrative review? systematic review? In the second case it could be important to follow the PRISMA guidelines and to revise all the manuscript accordingly. However, even if the selected type is that of narrative review, please add the criteria for article selection (year of publication, type of article, content of the article, the databases consulted, etc) because the definition “the most recent literature” is too general. In fact, the authors reported articles from 2013 to 2022. Are there any kind of articles excluded from the review? Why? Please justify the exclusion or add the remaining articles.
Response: Thank you for your constructive suggestion. Due to a lack of solid clinical evidence from randomized controlled trials, we think it is currently difficult to conduct a systemic review on this topic. Therefore, we decided to perform a narrative review to discuss the latest advances in the use of oXiris hemofilter in patients with critical illness in our original manuscript. Now, we have added the inclusion and exclusion criteria for article selection in the Introduction Section and slightly changed the title and abstract of our manuscript. In brief, the PubMed and Web of Science databases were searched for articles published from inception to September 8, 2022 (updated on November 1), with an English-language restriction (Lines 22-26, 59-66). Meanwhile, case report studies were excluded from the analysis due to their weak level of evidence and potential bias. All the revisions to the manuscript have been marked up using the “Track Changes” function in the revised manuscript.
Comment 3: Please remove from the review the articles available only in Chinese because they are not readable for all readers. It is important that the articles used for the review are widely available for consultation.
Response: Thank you for your constructive suggestion. We have removed all the articles published in Chinese by restricting the publication language to English only in the revised manuscript. (See revised Table 1)
Comment 4: Please add a table which summarizes the studies of oXiris in Covid19 patients (like what has already been done in table 1 for studies in septic patients).
Response: Thank you for your scientific comments and constructive suggestions. A table that summarizes the results of studies evaluating oXiris-CRRT in COVID-19 patients has been added to the revised manuscript, as shown in Table 2. (See revised Table 2)
Comment 5: In the introduction lines 51 and 52 the authors stated that “the therapeutic effect of oXiris-CRRT … has never been summarized in previous publication”: this is not fully correct. In literature there are two reviews available on Pubmed (Renal Replacement Techniques in Septic Shock. Hellman T, Uusalo P, Järvisalo MJ. Int J Mol Sci. 2021 Sep 23;22(19):10238. doi: 10.3390/ijms221910238. PMID: 34638575 and A Review of Extracorporeal Blood Purification Techniques for the Treatment of Critically Ill Coronavirus Disease 2019 Patients. Niazi NS, Nassar TI, Stewart IJ, Honore PM, Sharma K, Chung KK. ASAIO J. 2022 Apr 12. doi: 10.1097/MAT.0000000000001761. Online ahead of print. PMID: 35417433) that summarize the state of art in patients with sepsis and with Covid19 and cite the filter and briefly reported the studies available. I think it would be better to add these reviews in bibliography and to correct the statement in the introduction (for example: it is the first to focus on oXiris while others…)
Response: Thank you for your scientific comments and constructive suggestions. We have modified our statement in this section to avoid misleading information. It now reads as follows: “Although two recent publications have generally discussed the use of oXiris and other hemofilters in patients with sepsis or COVID-19, respectively [7,8], they unfortunately failed to address the safety concerns when delivering oXiris-CRRT sessions. With the increasing use of oXiris-CRRT in both septic and severe COVID-19 patients during the COVID-19 pandemic, it has become available and important to summarize the latest clinical evidence to investigate the efficacy and safety of oXiris-CRRT in critically ill patients through an updated narrative review that exclusively focuses on oXiris hemofilter.” (Line 51-58)
Comment 6: Table 1 summarizes the studies on septic patients: I think that it could be improve by adding two/three columns, one for the type of infection that affect patients in the studies (for example: gram negative or gram-positive infection, endocarditis, septic shock from … and so on) and one or two with the indication of the filters being compared in the study (if there is a comparison). Than summarize the statements in the “main findings”, removing the data of the other columns.
Response: Thank you for your scientific comments and constructive suggestions. We have added relevant columns to Table 1 to address your concern. (See revised Table 1)
Comment 7: The majority of studies in table 1 shows oXiris filter being compared with AN69ST: are there any study comparing oXiris with CytoSorb for example or other hemofilters? Why? Why not?
Response: Thank you for your scientific comments and constructive suggestions. We completely agree with your point that the therapeutic effect of the oXiris hemofilter should be compared with other hemofilters instead of its maternal AN69ST membrane. Unfortunately, we could hardly obtain any information on the comparison between the oXiris hemofilter and other hemofilters (including the Cytosorb hemofilter) through our literature search. This might be attributed to the fact that the initiation of oXiris-CRRT is usually considered when patients develop both AKI and septic shock. As you can see in Section 2, the AN69ST membrane can theoretically adsorb inflammatory cytokines by the negatively charged sulfonate groups in its bulk area. Therefore, the AN69ST membrane is widely used in sepsis-associated AKI patients in Chinese ICUs. In contrast, the use of Cytosorb standalone without CRRT in this patient population is unreasonable since Cytosorb cannot provide renal replacement therapy. Furthermore, the use of Cytosorb is still not approved by the Chinese Food and Drug Administration in mainland China, which unfortunately leads to a lack of relevant clinical evidence. We have also added some discussion to the Conclusion section to address this issue in the revised manuscript. (Line 392-395)
Comment 8: In the conclusion the authors stated “international multicentred registry studies”: how many registry studies are ongoing? Where? It would be valuable to give some information about them to allow other researchers to join them.
Response: Thank you for your scientific comment. Currently, there is an ongoing prospective international multicentered registry study (Global ARRT International registry, NCT03807414) that aims to identify a cluster of critically ill patients who mostly benefit from extracorporeal blood purification therapies with oXiris membranes. We have added more discussion to address this issue in our revised manuscript. (Line 363-365)
Comment 9: In the conclusion the authors stated that randomized controlled trials are desperately needed and they mention only an ongoing trial in the article (line 227-228). However, on Clinicaltrials.gov there are many different studies ongoing on oXiris filter: it would be interesting to add a paragraph, before conclusion, that summarize those studies, their aims, the expected term and findings etc., so giving the reader an idea about what is in progress.
Response: Thank you for your scientific comment and constructive suggestion. We have added a paragraph to Summarize the information of ongoing clinical studies evaluating oXiris-CRRT in critically ill patients in our revised manuscript. (See revised Table 3)
Comment 10: Table 1: French is a language; the country is France: please amend.
Response: Thank you for your scientific comment. We have corrected this mistake in Table 1. (See revised Table 1)

Reviewer 3 Report
This is a narrative review of utility of a PEI coated AN69ST filter in CRRT . This details the theoretical construct of the use of this filter along with review of studies. Studies are mostly retrospective with some prospective or RCT with surrogate outcomes. The review needs to be organised to provide a balance of benefits and harms of the filter. Studies need to organised into animal/invitro/retrospective/prospective/RCTs
Author Response
Comment: This is a narrative review of utility of a PEI coated AN69ST filter in CRRT. This details the theoretical construct of the use of this filter along with review of studies. Studies are mostly retrospective with some prospective or RCT with surrogate outcomes. The review needs to be organised to provide a balance of benefits and harms of the filter. Studies need to organised into animal/invitro/retrospective/prospective/RCTs.
Response: Thank you for your scientific comments and constructive suggestions to further improve the quality of our manuscript. The manuscript has now been thoroughly revised according to the comments from you and the other reviewers. We have tried to modify the subtitle of Section 3 to organize the included studies into preclinical and clinical studies and moved the discussion on the study that compared the removal of endotoxin and cytokines between the oXiris hemofilter and AN69ST hemofilter in septic patients by Broman et al to Section 3.3. (See lines 13-143) However, due to the lack of animal studies and RCTs in this field, we think it is difficult to mechanically elaborate the results of animal/in vitro/retrospective/prospective/RCTs one by one. All the revisions to the manuscript have been marked up using the “Track Changes” function in the revised manuscript.

Reviewer 4 Report
The authors present an interesting review on the hemofilter oXiris in critically ill patients, focusing in SARSCoV-2 patients. The review encompasses several issues regarding septic the structure of the filter, AKI, septic shock physiology, recent data on SARSCOV-2 and technical issues regarding coagulation and drug clearance.
Despite being reasonably organized, it can be improved and there are some important issues that should be addressed:
1) Line 180 – “The decision to initiate oXiris-CRRT is usually 180 driven by clinical judgment based on a combination of clinical signs, such as hemodynamic instability, microcirculatory dysfunction, MODS, and laboratory parameters, such as elevated serum levels of procalcitonin, IL-6, and lactate”. It is also important to refer the timing to dialysis (early or late start) and its impact on clinical outcomes as well as the advised dialysis dosing in AKI and septic shock patients
2) Section 4.1 should be more complete regarding the cytokine storm in SARSCOV-2 and which patients are more suitable to develop AKI. Also, the increased rate of thrombotic events in SARSCOV-2 should also be addressed since it might influence anti coagulation strategies.
3) An interesting study recently published showed that oXiris hemofilter is not effective in patients without AKI (1). This aspect should also be discussed.
I would also propose to present some issues in form of graphics and present a table with the studies in SARSCOV-2.
Author Response
Comment 1: The authors present an interesting review on the hemofilter oXiris in critically ill patients, focusing in SARSCoV-2 patients. The review encompasses several issues regarding septic the structure of the filter, AKI, septic shock physiology, recent data on SARSCOV-2 and technical issues regarding coagulation and drug clearance. Despite being reasonably organized, it can be improved and there are some important issues that should be addressed.
Response: Thank you for your scientific comments and constructive suggestions to further improve the quality of our manuscript. The manuscript has now been thoroughly revised according to the comments from you and the other reviewers. Please find below a list of our detailed responses to your comments and corresponding changes in the revised manuscript. All the revisions to the manuscript have been marked up using the “Track Changes” function in the revised manuscript.
Comment 2: Line 180 – “The decision to initiate oXiris-CRRT is usually driven by clinical judgment based on a combination of clinical signs, such as hemodynamic instability, microcirculatory dysfunction, MODS, and laboratory parameters, such as elevated serum levels of procalcitonin, IL-6, and lactate”. It is also important to refer the timing to dialysis (early or late start) and its impact on clinical outcomes as well as the advised dialysis dosing in AKI and septic shock patients.
Response: Thank you for your scientific comments and constructive suggestions. We have added more discussion on the timing to dialysis (Lines 199-211) and dialysis dose (Lines 219-224) in AKI and septic shock patients.
Comment 3: Section 4.1 should be more complete regarding the cytokine storm in SARSCOV-2 and which patients are more suitable to develop AKI. Also, the increased rate of thrombotic events in SARSCOV-2 should also be addressed since it might influence anti coagulation strategies.
Response: Thank you for your scientific comments and constructive suggestions. We have added more discussion of the cytokine storm and increased thrombotic events in COVID-19 patients to address your concern. We also discussed potential risk factors for developing AKI by analyzing the results of a recent multicenter cohort study (STOP-COVID study) across the USA. (Line 241-261)
Comment 4: An interesting study recently published showed that oXiris hemofilter is not effective in patients without AKI (1). This aspect should also be discussed.
Response: Thank you for your scientific comments and constructive suggestion. In a prospective nonblind randomized controlled study (10.3389/fphar.2022.817793), Kang K et al found that CRRT with an oXiris filter may not effectively alleviate cytokine storms in non-AKI patients with severe and critical COVID-19. The author thus suggested that the use of oXiris hemofilters outside of COVID-19 patients with AKI should be considered with caution to avoid increasing the unnecessary burden on society and individuals. Therefore, we have added a discussion on this issue by analyzing the results of the study by Kang K et al in the revised manuscript. (Line 270-278)
Comment 5: I would also propose to present some issues in form of graphics and present a table with the studies in SARSCOV-2.
Response: Thank you for your scientific comments and constructive suggestions. We have added a figure to show the readers a scheme of oXiris-CRRT and key messages for the delivery of oXiris-CRRT recommended by the Asia Pacific expert consensus. Furthermore, we have also added a table to summarize the results from recent clinical studies evaluating the use of oXiris-CRRT in critically ill COVID-19 patients. (see revised Figure 1 and Table 2)

Round 2
Reviewer 2 Report
I thank the authors for their answers that addressed all the previous concerns.
There is only one minor correction to do: line 314 "Gianluca et al", the correct citation of reference 50 is "Villa et al" because Villa is the surname and Gianluca the name. Please amend.
Author Response
Thank you for your scientific suggestion. We have corrected this mistake in our revised manuscript.